# Effects of Elevating Zinc Supplementation on the Health and Production Parameters of High-Producing Dairy Cows

**DOI:** 10.3390/ani14030395

**Published:** 2024-01-25

**Authors:** Maria Oconitrillo, Janaka Wickramasinghe, Shedrack Omale, Donald Beitz, Ranga Appuhamy

**Affiliations:** Department of Animal Science, Iowa State University, Ames, IA 50011, USA; mariaoh@iastate.edu (M.O.); janaka@iastate.edu (J.W.); shedrack@iastate.edu (S.O.); dcbeitz@iastate.edu (D.B.)

**Keywords:** antioxidant, lactation persistency, organic zinc, somatic cell count

## Abstract

**Simple Summary:**

Zinc (Zn) is an essential trace mineral required for milk production and maintaining the optimal health of dairy cows. The current nutrient requirement models (e.g., NASEM (2021)) predict the dietary Zn requirement of dairy cows to be 60 mg/kg of dry matter (DM), based on the milk production and dry matter intake (DMI) of cows. North American farmers, however, supplement Zn at a higher rate of 76 mg/kg of DM, and a meta-analysis demonstrated that even higher Zn supplementation at 100 mg/kg of DM could improve udder health. In this study, increasing the dietary Zn supplementation from 76 to around 97 mg/kg of DM decreased the DMI by about 1.0 kg/d, but it improved the post-peak milk yield, which was associated with an increased milk protein yield and decreased somatic cell count in milk. Overall, the data highlight the possibility of improving the udder health and milk yield persistency by increasing the organic Zn supplementation well above the current recommendations. Because milk yield persistence and low somatic cell counts have critical implications for economic viability and animal well-being, Zn supplementation will directly affect the overall sustainability of dairy production.

**Abstract:**

This study’s objective was to determine the effects of increasing the dietary added zinc (**Zn**) on the milk production, milk somatic cell count (**SCC**), and immunoglobulin and antioxidant marker concentrations in the blood of dairy cows. Twelve Holstein cows (67 ± 2.5 days in milk) were assigned randomly to (1) a diet containing Zn–methionine at 76 mg/kg of DM (**CTL**) or (2) CTL top-dressed with about 21 mg/kg of DM extra Zn–methionine (**+Zn**) for 70 d. The concentrations of reduced (**GSH**) and oxidized (**GSSG**) glutathione, malondialdehyde (**MDA**), catalase (**CAT**), superoxide dismutase (**SOD**), and immunoglobulins in the blood were measured on d 0, 35, and 70. Compared to CTL, +Zn decreased the dry matter intake (**DMI**) throughout the trial and the milk yield (**MY**) during the first phase of feeding (0–35 d). It, however, increased the milk yield during the last phase (36–70 d). The +Zn tended to have lower and greater milk protein yields than CTL during the first and last feeding phases, respectively. The +Zn tended to decrease the SCC and was associated with lower plasma GSH: GSSG and lower serum SOD concentrations relative to CTL. The +Zn did not affect the immunoglobulins, MDA, or CAT. Despite the early DMI and MY reduction, the prolonged Zn–methionine supplementation at about 100 mg/kg of DM improved the milk yield, possibly as a result of the improved udder health of dairy cows.

## 1. Introduction

Zinc (**Zn**) is an essential trace mineral that regulates enzymes in numerous biochemical pathways in the body. Zinc is a critical component of innate and adaptive immune responses [1,2,3], and it is required to maintain the thymus, where T cells undergo maturation [4,5]. Therefore, Zn has a positive impact on the production of antibodies such as immunoglobulin A (**IgA**), G (**IgG**), and M (**IgM**). Dietary Zn deficiency and the associated immunoglobulin deficiency can increase mastitis incidences in dairy cows [6]. Another mechanism by which Zn can decrease intramammary infection rates is by enhancing teat canal keratinization [7,8]. Moreover, Zn is an integral component of the body’s antioxidant system that alleviates oxidative stress by neutralizing reactive oxygen species such as superoxide anions (**O_2_^−^**) and hydrogen peroxide (**H_2_O_2_**). Zinc enhances the catalytic activity of the superoxide dismutase enzyme (**SOD**) that converts highly potent O_2_^−^ into less potent H_2_O_2_ [9]. Additionally, Zn enhances the activity of the glutathione peroxidase that catalyzes the oxidation of glutathione (**GSH**) into glutathione disulfide (**GSSG**) while neutralizing the H_2_O_2_ into H_2_O and O_2_ [10,11]. The significance of Zn in the immune and antioxidant functions is poorly understood in high-producing dairy cows and the relevant data are scarce in the published literature despite the increased likelihood of experiencing immune challenges and oxidative stress [12].

Organic forms of Zn have garnered significant attention in the last two decades as a result of their high bioavailability [13]. Among the organic forms, Zn–amino acid complexes (**Zn**–**AA**), such as Zn–methionine complexes (**Zn**–**Met**) are frequently used in dairy cow diets. Regardless of the Zn source, the NASEM [14] recommends 60 mg of total dietary Zn in one kg of dietary DM for a dairy cow weighing 650 kg and producing 50 kg of milk daily. Farmers, however, seem to supplement Zn at a much higher rate, as a commercial dairy farm survey conducted by Duplessis et al. [15] showed an average Zn supplementation at 76 mg/kg of DM. Furthermore, through a meta-analysis, Kellogg et al. [16] demonstrated that increasing the Zn–Met supplementation above the current recommendation could improve the udder health of dairy cows, as indicated by decreased somatic cell counts (**SCCs**) in milk. In support, increasing the added Zn up to 90 mg per kg of DM decreased the inflammation and rectal temperature and increased the weight gain of beef cattle fed ractopamine hydrochloride [17] or challenged with bovine rhinotracheitis virus [18]. Moreover, Zhao et al. [19] showed that increasing the organic Zn concentration in the diet from 50 to 100 mg/kg DM increased the concentrations of serum Zn, GSH, and SOD in lactating dairy cows. Based on the literature, we hypothesized that elevating the organic Zn supplementation up to nearly 100 mg/kg of DM would improve the health of dairy cows and, thus, production performances. The study’s objective was to determine the effects of increasing the dietary Zn–Met concentration from 76 to 97 mg/kg DM on the SCC and immunoglobulin, antioxidant enzymes, and oxidative stress marker concentrations in the blood, as well as the effects on the feed intake and milk production performances of high-producing dairy cows.

## 2. Materials and Methods

### 2.1. Animals and Treatments

All the animal procedures were approved by the Institutional Animal Care and Use Committee at Iowa State University (IACUC-20-104). The study was conducted from August to October 2020 at the Dairy Research and Teaching Farm at Iowa State University. Six primiparous and six multiparous Holstein dairy cows (628.0 ± 19.0 kg of body weight; 67.0 ± 2.5 days in milk) were blocked by parity and assigned randomly to one of two dietary treatments (*n* = 6 cows per treatment): (1) a basal total mixed ration (**CTL**) containing Zn–Met at 76 mg/kg of DM of Zn–Met or (2) CTL top-dressed with additional Zn–Met at around 20 mg/kg of DM (**+Zn**) for 70 d. Depending on the average baseline DMI of the +Zn cows added to a 10% safety margin (23.4 + 2.34 = 25.7 kg/d), 516 mg of total Zn–Met was top-dressed daily on the basal TMR delivered to every Zn+ cow to achieve the extra 20 mg/kg of DM supplementation throughout the 70 d trial. When adjusted for the variability of the DMI across the trial, the mean and the standard deviation (SD) of the Zn + Met supplementation were 20.9 and 2.5 mg/kg of DM, respectively. A statistical power analysis (power = 0.80 and α = 0.05) was conducted using SCC (SD = 20 × 10^3^/mL [20]), IgG (SD = 1.2 mg/mL [21]), and SOD (SD = 0.36 U/mL [22]) data to determine the sample size required to capture the statistical significance of effect sizes representing a 25% change from the mean. The analysis revealed the sample size of 6 cows per treatment would be adequate to capture the statistical significance of a 15, 10, and 25% change from the means of SCC, IgG, and SOD, respectively.

### 2.2. Feeding and Sample Collection

The animals were housed in a free-stall barn and had access to clean drinking water throughout the study period. The cows were fed ad libitum (110% of the previous day’s intake) twice daily (0600 and 1500 h) using the Calan Broadbent Feeding System (American Calan Inc., Northwood, NH, USA). The +Zn was top-dressed on the basal total mixed ration delivered at 06:00 h. The cows were milked twice daily (1100 and 2300 h), and the milk yield was recorded daily. The cows were trained on the Calan feeding system for 6 d before recording the baseline measurements for 4 d. The baseline records included the dry matter intake (**DMI**), body weight, body condition score (BCS), and milk production parameters. The blood was collected via jugular venipuncture into vacutainer tubes (Becton, Dickinson and Company, Franklin Lakes, NJ, USA) to obtain plasma (tubes with K2EDTA) and serum (tubes without an anticoagulant) at 16:00 h on the last day of the baseline measurement period (d 0). Following the baseline measurements, the top-dressing +Zn was begun and continued over the next 70 d. Samples of the basal diet were collected six times a week and composited into weekly samples for nutrient composition analysis (Dairyland Laboratories Inc., Arcadia, WI, USA; Table 1). Orts samples were collected six times a week and composited into weekly samples for dry matter content analysis via oven-drying (60 °C for 48 h). Milk samples were collected from both milking sessions once every two weeks and stored at 4 °C with a preservative (Bronopol tablet; D & F Control Systems Inc., San Ramon, CA, USA) until being analyzed for the milk composition. Milk samples were analyzed for the true protein, fat, lactose, and SCC (Dairy Lab Services, Dubuque, IA, USA). Milk samples were collected from the morning and evening milking sessions on d 0, 35, and 70, composited proportionate to the milk weights at each session within the day, and kept frozen at −20 °C until being analyzed for the Zn concentrations. Jugular blood was drawn again to obtain plasma and serum on d 35, and 70. In all cases, the blood collected for serum was allowed to clot at room temperature for about an hour before centrifugation. After centrifugation at 1500× *g* and 4 °C for 15 min, serum and plasma were harvested and stored at −20 °C until further analysis. An extra blood sample was obtained into a 5 mL vacutainer tube (K2EDTA; Becton, Dickinson and Company, Franklin Lakes, NJ, USA), stored at 4 °C, and sent to the Veterinary Diagnostic Laboratory at Iowa State University (Ames, IA, USA) for complete blood count analysis within 24 of blood having been collected.

### 2.3. Analysis of Antioxidants and Immunoglobulins in Blood

The plasma concentrations of reduced (**GSH**) and oxidized (**GSSG**) glutathione, malondialdehyde (**MDA**), and serum concentrations of catalase (**CAT**), superoxide dismutase (**SOD**), and immunoglobulins including IgG, IgA, and IgM were analyzed using commercial kits according to the instructions provided by the manufacturers (GSH and GSSG: Arbor Assays, Ann Arbor, MI, USA; MDA: Cayman Chemical, Ann Arbor, MI, USA; CAT: Invitrogen, Carlsbad, CA, USA; SOD: Cayman Chemical, Ann Arbor, MI, USA; IgG, IgA, IgM: Bethyl Laboratories, Montgomery, TX, USA). The dilution rates used in the analyses, except the MDA analysis requiring no dilution, were 1:5 (GSH and GSSG), 1:10 (CAT), 1:5 (SOD), 1: 1000 (IgM), 1: 2500 (IgG), and 1:500 (IgA). Each analysis included 36 samples representing 12 cows at d 0, 35, and 70 of the study. Each sample was analyzed in duplicate, and all the samples were analyzed in a single assay (one 96 well plate) using a microplate spectrophotometer (BioTek Eon, BioTek Instruments, Inc., Winooski, VT, USA) with data analysis software (BioTek gen5, version 2.03). The analysis was, however, repeated for a given sample when the coefficient of variation across the duplicates was >10%. 

### 2.4. Analysis of Zinc Concentrations in Milk and Serum

The milk and the serum samples were analyzed using inductively coupled plasma mass spectrometry (ICP-MS) in the Veterinary Diagnostic Laboratory at Iowa State University (Ames, IA, USA). 

***Milk***. A 0.25 mL portion of each milk sample was transferred to a 15 mL tube, and 0.25 mL of 70% nitric acid was added to it. The samples were digested at 60 °C for 1.0 h and then brought up to 5.0 mL using 1.0% nitric acid. The samples were then vortexed and filtered through a 0.45 µm filter disc. The filtered samples were injected into the ICP-MS (PlasmaQuant MS Elite^®^, Analytik Jena, Jena, Germany).

***Serum***. The samples were diluted at 1:20 via mixing a 0.25 mL portion of each serum sample with 4.75 mL of 1.0% nitric acid in a 15 mL tube. The diluted samples were vortexed and injected into the ICP-MS (PlasmaQuant MS Elite^®^, Analytik Jena, Jena, Germany).

### 2.5. Calculations and Statistical Analysis

The protein, fat, and lactose yields were calculated by multiplying the milk yield by the corresponding component concentration. The feed efficiency (**FE**) was calculated by taking the ratio between the milk yield (kg/d) and DMI (kg/d). The energy-corrected milk (**ECM**, kg/d) was calculated using the following equation [23]: ECM = [(0.327 × milk yield, kg/d) + (12.95 × fat yield, kg/d) + (7.65 × protein yield, kg/d)]

The ratio between the reduced and oxidized glutathione (**GSH: GSSG**) was calculated to describe the overall oxidative stress status [24].

***Statistical analysis***. All the data except the SCC were analyzed using the MIXED procedure of SAS (version 9.4, SAS Institute Inc., Cary, NC). The statistical model included the fixed effects of the treatment (CTL or +Zn), treatment feeding phase (d 0 to 35 or d 35 to 70) or sampling time point (d 35 or d 70), parity of the cows (primiparous or multiparous), treatment × time interaction, and covariate effects of the baseline measurements. The random effect was the cow nested in the treatment. The same model was applied using PROC GLIMMIX with a Poisson distribution for the analysis of the SCC. All the data are expressed as the least squares means and the statistical significance of the effects was declared at *p* ≤ 0.05. The tendencies were discussed at 0.05 < *p* < 0.10. 

## 3. Results and Discussion

Sustainable dairy production equally demands improvements in milk production as well as the well-being of dairy cows. Some trace minerals are vital for immune defense and alleviating physiological stresses, and Zn is recognized as one such mineral [25,26]. Zinc enhances the activity of many enzymes required for nutrient metabolism, antibody production, and neutralizing reactive oxygen species. Improving the bioavailability and ensuring optimal immune function and redox status are principal goals of the trace mineral nutrition of dairy cattle [26]. In that regard, Zn–AA has become popular among Zn supplementations in commercial farms, which have been surveyed to be at 76 mg/kg of DM [15]. However, current recommendations concerning the Zn requirements encompass only the maintenance and production requirements, and some of the literature supports higher Zn supplementation to improve cow health. Therefore, an elevated Zn–methionine supplementation (76 to 97 mg/kg of DM) in dairy cow diets was investigated for the SSC, blood cell count, and blood immunoglobulin and antioxidant marker concentrations, as well as the feed intake and production performances, of high-producing dairy cows in this study. 

### 3.1. Feed Intake and Production Performance

The least squares means and statistical significance of the treatment effects of the DMI, milk production parameters, and body weight during the first (0 to 35 d) and the second phases (35 to 70 d) of the feeding treatments are given in Table 2. Figure 1 depicts the weekly means of the DMI and milk yield. The first and second phases of the feeding treatments corresponded to 77 to 112 and 113 to 147 DIM, respectively, demarcating the transition from the early to mid-lactation stage. The elevated supplementation of Zn–Met decreased the DMI by 1.20 kg/d compared to the DMI of CTL throughout the study, when adjusted statistically for the differences in the baseline DMI (*p* < 0.01, Figure 1 and Table 1). In support, Allahyari et al. [27] observed increased blood concentrations of anorexigenic hormones, such as leptin and insulin, in Holstein cows when the added Zn in the diet increased from 110 to 250 mg/kg of DM. The Zn supplementation affected the milk yield in a time-dependent manner, as indicated by the treatment × time interaction in Table 2 (*p* < 0.01). The milk yield of +Zn was 2.00 kg less than CTL in the first feeding phase (*p* < 0.01), but +Zn increased the milk yield by 1.25 kg compared to CTL in the second phase (*p* = 0.02). The treatment × time interaction tended to affect the lactose yield, the major driver of the milk volume, in the same way (*p* = 0.07). The reasons behind the +Zn × time interaction on the milk yield are not fully understood. Nevertheless, the second half of the study overlaps with 113 to 147 days in milk where the milk yield gradually decreases after achieving the peak lactation as a result of decreasing the secretory cell numbers in the mammary glands. The increased milk yield in response to +Zn could be a result of the mammary glands being able to sustain the secretory capacity in mid-lactation, as Zn can stimulate cellular signaling pathways (e.g., JAK2/STAT5, and MAPK), enhancing the proliferation and survival of mammary epithelial cells [28]. Zinc can modulate lactose synthesis, depending on the Mn^2+^ concentration in mammary epithelial cells, such that Zn can decrease lactose synthase activity at high intracellular Mn^2+^ concentrations [29]. Because Zn and manganese can share the same cell membrane transporters in the body [30], it can be speculated that the increased serum Zn of +Zn during the last rather than the first feeding phase competitively inhibited the Mn^+2^ uptake by mammary epithelial cells to increase the lactose and thus milk yields. Nevertheless, data supporting such inhibition, particularly in mammary cells, are scarce in the published literature. 

The milk protein and fat concentrations and milk fat yield were not affected by +Zn or the +Zn × time interaction (*p* > 0.60). Similar to what was observed for the milk and lactose yields, +Zn tended to affect the milk protein yield differently in the first and second feeding phases (*p* = 0.07). During the first 35 d, the milk protein yield of CTL was 50 g greater than that of +Zn, but it increased by 40 g in response to +Zn compared to CTL during the next 35 d. Elevating the Zn supplementation from 76 to 97 mg/kg of the DM tended to decrease the SCC (*p* = 0.07) throughout the 70 d (Table 2). The reduction was more prominent during the last 35 d compared to the first 35 d (*p*-values = 0.07 vs. 0.30) when adjusted for the baseline SCC. Even though the SCC may not be as sensitive as bacterial culture results in identifying udder infections [31], its relationship with the total viable bacterial counts in milk makes the SCC an effective udder health indicator [32]. Nyman et al. [33] demonstrated that an SCC as low as 74 × 10^3^ cells/mL was related to intra-mammary infections despite the SCC of 200 × 10^3^ cells/mL being usually used as the cutoff to determine subclinical mastitis in dairy cows. Somatic cells in milk contain primarily white blood cells and cells sloughed from the mammary epithelium. The lack of differences in the blood cell counts (*p* > 0.25) presented in Table 3 suggest the SCC decrease could be predominantly a result of a mammary cell loss decrease. Zinc could decrease the contribution of mammary epithelial cells to the SCC, as it improves mammary epithelial integrity [34,35]. Therefore, the SCC results support the aforementioned notion that +Zn could potentially improve the milk yield persistency by mitigating the post-peak mammary epithelial cell loss. The +Zn did not affect the ECM (*p* = 0.55), FE (milk yield: DMI, *p* = 0.19), body weight (*p* = 0.12), or BCS (*p* = 0.39, Table 2). Moreover, +Zn did not modify the other hematological parameters, such as the hemoglobin concentration, hematocrit %, mean corpuscular volume, and mean platelet volume (*p* > 0.1, Table 3).

### 3.2. Zinc Concentrations in Blood Serum and Milk

The zinc concentrations in the milk and blood serum can be found in Table 2 and Table 4, respectively. The zinc concentration in the milk of +Zn tended to be greater than that of CTL throughout the study (4.48 vs. 4.06 ppm, *p* = 0.07). The milk Zn concentrations across the treatments were within the reference ranges in the literature [36,37]. The data in Spolders et al. [38] provide a reference range of 0.71–1.35 ppm for the serum Zn concentrations in lactating dairy cows, and the concentrations of this study are within that range. There was a significant interaction between the treatment and the sampling time for serum Zn concentration (*p* = 0.04, Table 4). At d 35 of feeding, CTL and +Zn had similar (*p* = 0.47) serum Zn concentrations at 0.76 ppm, but +Zn had a higher serum Zn concentration than CTL (1.06 vs. 0.82 ppm, *p* = 0.05) at d 70. The serum Zn concentration of +Zn being similar to that of CTL at d 35 despite the elevated Zn supply from the diet and the decreased milk Zn efflux was unexpected. One potential reason why the serum Zn concentration could reflect the Zn supplementation only at d 70 would be related to an increased immune activation [39] and, thus, an increased Zn sequestration by the immune system [40] overriding the effects of the dietary Zn supply and milk Zn efflux on blood Zn homeostasis in early lactation compared to the mid-lactation in dairy cows. 

### 3.3. Immunoglobin and Antioxidant Marker Concentrations in Blood

The serum immunoglobulin concentrations and the concentrations of antioxidant or oxidative stress markers are presented in Table 4. The +Zn did not affect the concentrations of immunoglobulins, such as IgA, IgM, and IgG (*p* > 0.25). On the contrary, Zn deficiency impairs B cell development and, thus, the production of antibodies [41,42,43,44]. Moreover, dietary Zn supplements increase the blood immunoglobulin concentrations of calves, broilers, and piglets [45,46,47]. Chen et al. [48] showed negligible changes in the serum IgA, IgG, or IgM concentrations in response to Zn supplementations at 20, 40, or 60 mg/kg of DM in dairy cows. On the other hand, Chandra et al. [49] observed higher plasma IgG concentrations in pre- and post-partum cows in response to Zn supplemented at 110 mg/kg of DM compared to a 50 mg/kg supplement. Perhaps the Zn–Met supplementation at 76 mg/kg of DM (CTL) alone could sustain the optimal immunoglobulin production in cows.

**Table 2 animals-14-00395-t002:** The least squares means and the treatment effects of the feed intake, milk production, feed efficiency, and growth parameters of lactating dairy cows (77 to 147 days in milk) consuming dietary added Zn (Zn–methionine) at 76 mg/kg DM (CTL) or 97 mg/kg DM (+Zn) ^1^.

Variable ^2^	1 to 35 d		36 to 70 d		SEM	*p*-Value		
	CTL	+Zn	CTL	+Zn		Trt	Period	Trt × Period
DMI, kg/d	26.10	24.90	25.80	24.60	0.32	<0.01	0.31	0.96
Milk yield, kg/d	42.00 ^a^	39.81 ^b^	39.89 ^b^	40.90 ^c^	0.34	0.17	0.12	<0.01
Milk composition								
Protein, %	3.13	3.11	3.24	3.22	0.07	0.83	<0.01	0.92
Fat, %	4.34	4.28	4.47	4.33	0.16	0.63	0.34	0.68
Lactose, %	4.79	4.76	4.69	4.71	0.03	0.92	<0.01	0.24
Protein yield, kg/d	1.30	1.25	1.27	1.31	0.06	0.99	0.64	0.07
Fat yield, kg/d	1.68	1.59	1.67	1.65	0.10	0.66	1.71	0.62
Lactose yield, kg/d	2.00	1.96	1.85	1.97	0.09	0.75	0.11	0.07
SCC, ×10^3^/mL	457	252	251	69	112	0.06	0.03	0.38
Zinc, ppm	4.06	4.30	4.05	4.65	0.22	0.07	0.45	0.40
ECM, kg/d ^3^	46.61	43.04	44.81	44.70	2.52	0.55	0.96	0.27
Milk yield: DMI	1.61	1.64	1.57	1.67	0.04	0.19	0.89	0.46
Body weight, kg	663	646	650	643	7.20	0.12	0.25	0.45
BCS	2.50	2.56	2.64	2.68	0.06	0.39	0.05	0.84

^1^ The least squares means and treatment effects are adjusted for the baseline measurements. ^2^ DMI = dry matter intake, SCC = somatic cell count, ECM = energy corrected milk, and BCS = body condition score. ^3^ ECM = [(0.327 × Milk yield, kg/d) + (12.95 × fat yield, kg/d) + (7.65 × protein yield, kg/d)]. ^abc^ Different superscripts in the same row indicate different least squares means (*p* < 0.05).

**Table 3 animals-14-00395-t003:** The least squares means and the treatment effects of the blood cell count and other hematology parameters of lactating dairy cows (77 to 147 days in milk) consuming dietary added Zn (Zn–methionine) at 76 mg/kg DM (CTL) or 97 mg/kg DM (+Zn) ^1^.

Variable	35 d		70 d		SEM	*p*-Value		
	CTL	+Zn	CTL	+Zn		Trt	Period	Trt × Period
White blood cells, ×10^3^/µL	9.48	10.05	10.92	11.02	0.92	0.74	0.20	0.80
Red blood cells, ×10^3^/µL	6.36	6.71	6.60	6.90	0.28	0.40	0.22	0.89
Hemoglobin, g/dL	10.65	11.15	11.25	11.75	0.41	0.40	0.03	0.99
Hematocrit, %	29.19	30.35	30.76	31.87	1.19	0.50	0.07	0.97
MCV, µm^3^	46.36	44.82	46.94	45.74	0.69	0.19	<0.01	0.20
Platelets, ×10^3^/µL	355.40	332.60	358.90	343.20	35.81	0.67	0.78	0.89
Mean platelet volume, µm^3^	6.95	5.89	6.48	6.97	0.71	0.69	0.67	0.30
Neutrophils, ×10^3^/µL	4.69	4.80	6.24	5.40	0.97	0.72	0.28	0.63
Lymphocyte, ×10^3^/µL	3.58	4.41	3.55	4.66	0.59	0.30	0.59	0.48
Monocyte, ×10^3^/µL	0.46	0.57	0.50	0.61	0.09	0.29	0.62	0.99
Eosinophils×10^3^/µL	0.39	0.32	0.18	0.34	0.11	0.71	0.43	0.33
Basophils, ×10^3^/µL	0.11	0.13	0.13	0.15	0.02	0.32	0.28	0.99

^1^ The least squares means and treatment effects are adjusted for the baseline values.

**Table 4 animals-14-00395-t004:** The least squares means and the treatment effects of the serum Zn, immunoglobulin, and oxidative markers of lactating dairy cows (77 to 147 days in milk) consuming dietary added Zn (Zn–methionine) at 76 mg/kg DM (CTL) or 97 mg/kg DM (+Zn) ^1^.

Variable	35 d		70 d		SEM	*p*-Value		
	CTL	+Zn	CTL	+Zn		Trt	Period	Trt × Period
Serum Zn, ppm	0.80 ^b^	0.71 ^b^	0.82 ^b^	1.06 ^a^	0.08	0.38	0.02	0.04
Immunoglobulins								
IgA, mg/mL	0.12	0.12	0.10	0.11	0.01	0.76	0.09	0.11
IgG, mg/mL	27.51	28.88	27.90	22.80	1.75	0.27	0.10	0.11
IgM, mg/mL	1.53	1.53	1.40	1.48	0.19	0.87	0.57	0.80
Antioxidant markers ^2^								
CAT, U/mL	3.60	3.95	4.73	3.61	0.69	0.62	0.55	0.28
SOD, U/mL	4.53	4.79	7.70	4.25	1.03	0.18	0.21	0.09
MDA, µM	7.17	9.05	8.64	8.63	1.75	0.68	0.71	0.51
GSH, µM	2.47	2.09	2.37	2.01	0.10	0.02	0.35	0.82
GSSG, µM	0.22 ^c^	0.40 ^a^	0.37 ^ab^	0.34 ^b^	0.04	0.09	0.15	<0.01
GSH: GSSG	12.00 ^b^	5.30 ^a^	6.40 ^a^	6.40 ^a^	0.95	<0.01	0.02	<0.01

^1^ The least squares means and treatment effects are adjusted for the baseline values. ^2^ Reduced (GSH) and oxidized (GSSG) glutathione, malondialdehyde (MDA), catalase (CAT), and superoxide dismutase (SOD). ^abc^ Different superscripts in the same row indicate different least squares means (*p* < 0.05).

Oxidative stress is a metabolic dysfunction that favors the oxidation of biomolecules such as DNA, protein, and lipids, contributing to the oxidative damage to cells and tissues [50]. The literature implies that Zn plays a role in protecting cells and tissues from oxidative damage in humans [51]. Such evidence is sparse for cows, even though the high-producing cows, in particular, experience significant oxidative stress associated with their high metabolic demands [52]. In this study, we examined the effect of Zn supplementation on the antioxidant capacity and the degree of oxidative stress in dairy cows by evaluating the concentrations of antioxidant enzymes, such as CAT and SOD, the concentration of MDA, a degradation product of lipid peroxidation, and GSH: GSSG in the blood. The results (Table 4) did not support any effects of +Zn on the CAT and MDA concentrations (*p* > 0.170). The Zn supplementation exhibited only a tendency to affect the serum SOD concentrations in a time-dependent manner, as indicated by the treatment × time interaction (*p* = 0.09). The +Zn decreased the GSH concentration (*p* = 0.02) and tended to increase the GSSG concentration (*p* = 0.09) compared to CTL on d 35. Consequently, the GSH: GSSG decreased in response to +Zn relative to CTL (*p* < 0.01) on d 35. The serum GSH: GSSG can indicate the degree of whole-body oxidative stress, with lowered values indicating elevated oxidative stress [24]. Therefore, the decreased GSH: GSSG on d 35 could correspond to elevated oxidative stress in response to +Zn [52]. Future investigations taking comprehensive approaches to determine the activity of different SOD species [53] and the activity of enzymes, such as glutathione peroxidase oxidizing GSH to GSSG [10,11] may help explain well what those biomarker changes mean. For instance, such analyses would decipher if the decreased GSH: GSSG reflects increased oxidative stress or a successful alleviation of oxidative stress in response to +Zn [54]. A post hoc power analysis comparable to the initial analysis revealed sample sizes of 7, 9, and 14 to capture the statistical significance (*p* < 0.05) of the observed effect sizes of the SCC, SOD, and IgG, respectively. Therefore, future investigations equipped with improved statistical power would enable capturing the true effects of Zn–Met supplementation on the immune and antioxidant system-related traits of dairy cows. 

## 4. Conclusions

The results highlight that elevating the dietary Zn–Met supplementation from 76.0 to 97.0 ± 2.5 mg/kg of DM over 70 d decreased the milk yield during the first half of the trial but was associated with increased milk yields during the second half, despite the DMI continuing to be lowered by about 1.0 kg/d throughout the trial. The first and second halves of the trial represented the transition from the early to mid-lactation stages. The improved milk yield seems to result from an improvement in the mammary secretory cell number and integrity, as reflected in the improved milk somatic cell counts, while the white blood cell counts remained unchanged in response to the Zn supplementation. The Zn supplementation did not affect the blood immunoglobulin concentrations and tended to change the concentrations of antioxidant capacity or oxidative stress markers in the blood. Comprehensive future investigations with improved statistical power are required to explain well the effects of Zn–Met supplementation on the production and the well-being of dairy cows.

## Figures and Tables

**Figure 1 animals-14-00395-f001:**
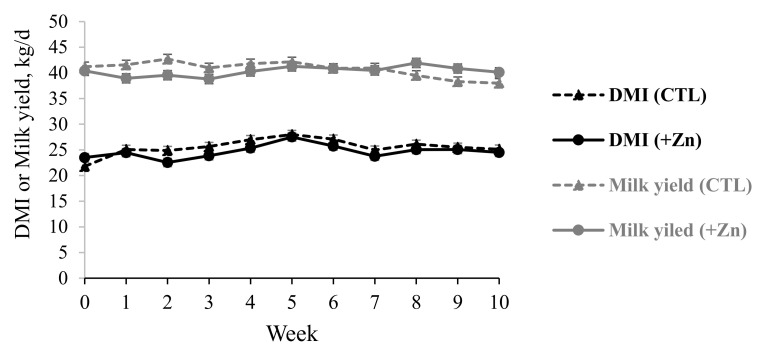
Weekly mean DMI (kg/d) and milk yield of cows supplemented with Zn–methionine at 76 mg/kg DM (CTL) or 97 mg/kg DM (+Zn).

**Table 1 animals-14-00395-t001:** Ingredient and nutrient composition of the basal diet.

Item	Value
*Ingredient composition, % of DM*	
Corn silage	42.73
Alfalfa hay	12.75
Ground corn	17.13
Corn gluten feed	8.65
Expeller soybean	4.59
Soybean meal	4.41
Straw	1.24
Concentrate premix ^1^	8.49
*Nutrient composition (% of DM, unless otherwise mentioned)*	
DM (% as-fed)	52.50
Starch	26.97
CP	15.33
NDF	31.23
ADF	20.50
Ether extract	4.05
Ca	0.94
P	0.38
Mg	0.34
K	1.51
Na	0.58
Fe (mg/kg DM)	375.33
Mn (mg/kg DM)	64.00
Zn-methionine (mg/kg DM)	76.00
Total Zn (mg/kg DM) ^2^	108.20
Cu (mg/kg DM)	14.00
NEL (Mcal/kg of DM)	1.65

^1^ Bypass soybean (19.2%), blood meal (13.9%), palm fat (13.3%), calcium carbonate (14.4%), sodium bicarbonate (11.3%), pork meat and bone meal (9.7%), sodium chloride (4.7%), soybean meal (3.1%), magnesium oxide (2.7%), trace mineral mix (2.3%), urea (2.5%), choice white grease (1.2%), yeast culture (0.7%), rumen-protected methionine (0.8%), organic chromium (0.1%), monensin (0.07%), biotin (0.04%), and Zn-methionine (0.02%). ^2^ Zn from Zn–methionine and the other ingredients. The latter was calculated from the tables (NRC, Rockville, MD, USA, 2001).

## Data Availability

All the datasets collected and analyzed during the current study are available from the corresponding author on fair request.

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
