# Peer review of "Effects of Elevating Zinc Supplementation on the Health and Production Parameters of High-Producing Dairy Cows"

_animals, 2024, doi:10.3390/ani14030395_

Round 1
Reviewer 1 Report
Comments and Suggestions for Authors
Peer review report on “Effects of Elevating Zinc Supplementation on Health and Pro-duction Parameters of High-producing Dairy Cows”.
General comments:
The use of essential trace minerals in animal feeding is a very interesting area and one that is worthy of experimental attention. Zinc, in addition to regulate several enzymes, is an important component of immune responses and antioxidant system. The authors researched the link between the increasing organic Zn concentration and health and production parameters of dairy cows. The opportunity to improve milk yield persistency by improving the udder health is always of great interest to both farmers and consumers.
The chapters of the manuscript are mostly fair and well structured, the paper is generally clear and well written, and the information obtained can deliver useful sights for the readers. However, I have some concerns and explain them in more detail below.
Minor comments:
Line 41: The first reference mentioned is (3). Where are references 1. and 2.?
Line 84: Were the cows in the same parity?
Line 87-91: The results of the power analysis is not clear, as the P value used to evaluate the results is 0.05 instead of 0.1. Please explain the discrepancy.
Line 90: statistical significance of 10, 15 and 25%
Line 101: What anticoagulant was used to separate the plasma?
Line 111: Please add full names of MUN and SCC
Table 1: Line 127, instead of "concentration", I would suggest "value"
Line 163: Analysis of Antioxidants and Immunoglobulins in Blood
Line 164-167: Please indicate the dilution ratio of the samples for the different parameters.
Line 174: The type and manufacturer of microplate reader is missing. Which software did you used to interpolate the sample concentrations from the standard curves?
Line 181: the speed and duration of centrifugation is missing
Table 3: The threshold of P value (P<0.05) is missing from the caption. There is a single sentence in the text that mentions Table 3 in relation to blood cell counts, but I could not find any textual evaluation of the other hematology parameters in the manuscript.
Author Response
Thank you for your edits, comments, and suggestions. We revised the manuscript by addressing all your comments. Revised texts are highlighted in yellow in the revised manuscript.
Reviewer 1
General comments:
The use of essential trace minerals in animal feeding is a very interesting area and one that is worthy of experimental attention. Zinc, in addition to regulate several enzymes, is an important component of immune responses and antioxidant system. The authors researched the link between the increasing organic Zn concentration and health and production parameters of dairy cows. The opportunity to improve milk yield persistency by improving the udder health is always of great interest to both farmers and consumers.
The chapters of the manuscript are mostly fair and well structured, the paper is generally clear and well written, and the information obtained can deliver useful sights for the readers. However, I have some concerns and explain them in more detail below.
Minor comments:
Line 41: The first reference mentioned is (3). Where are references 1. and 2?
AU: Corrected (Line 3)
Line 84: Were the cows in the same parity?
AU: Six primiparous and six multiparous Holstein dairy cows (Line 84)
Line 87-91: The results of the power analysis is not clear, as the P value used to evaluate the results is 0.05 instead of 0.1. Please explain the discrepancy.
AU: The section was revised as follows
A statistical power analysis (power = 0.80 and α = 0.05) was conducted using SCC (σ = 20 ×103/mL [20]) IgG (σ = 1.2 mg/mL [21]), and SOD (σ = 0.36 U/mL [22]) data to determine the sample size to capture the statistical significance of effect sizes representing a 25% change from the mean. The analysis revealed the sample size of 6 cows per treatment would be adequate to capture the statistical significance of 15, 10, and 25% change from the means of SCC, IgG, and SOD, respectively. (Lines 93-99)
Line 90: statistical significance of 10, 15 and 25%
AU: Please refer to the response above
Line 101: What anticoagulant was used to separate the plasma?
AU: The sentence was revised as follows
The blood was collected by jugular venipuncture into vacutainer tubes (Becton, Dickinson and Company, Franklin Lakes, NJ) to obtain plasma (tubes with K2EDTA) and serum (tubes without an anticoagulant). Lines 109-111
Line 111: Please add full names of MUN and SCC
AU: The SCC is defined in Line 66. The MUN was removed, as it was not included in the results.
Table 1: Line 127, instead of "concentration", I would suggest "value"
AU: Revised accordingly
Line 163: Analysis of Antioxidants and Immunoglobulins in Blood
AU: Revised accordingly (revised accordingly in line 170)
Line 164-167: Please indicate the dilution ratio of the samples for the different parameters.
AU: The dilution rates used in the analyses except the MDA analysis requiring no dilution were 1:5 (GSH and GSSG), 1:10 (CAT), 1:5 (SOD), 1: 1000 (IgM), 1: 2500 (IgG), and 1:500 (IgA). (Lines 177-179)
Line 174: The type and manufacturer of microplate reader is missing. Which software did you used to interpolate the sample concentrations from the standard curves?
AU: given as follows
The samples were analyzed in a single assay (one 96 well plate) by using a microplate spectrophotometer (BioTek Eon, Biotek Instruments, Inc., Winooski, Vermont, USA) with a data analysis software (BioTek gen5, version 2.03). Lines 181-183
Line 181: the speed and duration of centrifugation is missing
AU: When double-checked with the laboratory recently, we realized that no centrifugation was used in their procedure for our samples. Therefore, it was removed from the text in the revised manuscript.
Table 3: The threshold of P value (P<0.05) is missing from the caption.
Such a caption is not necessary, as none of the variables was associated with treatment x time interaction to indicate the differences (P<0.05) using the superscripts, a, b and c.
There is a single sentence in the text that mentions Table 3 in relation to blood cell counts, but I could not find any textual evaluation of the other hematology parameters in the manuscript.
AU: The other haematology parameters were described briefly as follows
Moreover, +Zn did not modify the other hematological parameters, such as hemoglobin concentration, hematocrit %, mean corpuscular volume, and mean platelet volume (P > 0.1, Table 3). Lines 283-285
Reviewer 2 Report
Comments and Suggestions for Authors
The manuscript evaluates the effects of increasing the dietary added zinc on milk production, milk somatic cell count, and immunoglobulin and antioxidant concentrations in the blood of dairy Holstein cows. The manuscript is well-designed and the measured parameters are enough to have a solid conclusion. However, some recommendations may help to improve the manuscript before publication.
1) There is little originality/novelty of the presented study in this aspect.
2) Although the experimental design and analysis of the data seem correct, I would like to check the SAS codes.
3) Why do you have the treatment feeding phase (d 0 to 35 or d 35 to 70)?
Author Response
Thank you for your edits, comments, and suggestions. We revised the manuscript by addressing all your comments. Revised texts are highlighted in yellow in the revised manuscript. Also, the SAS codes you had asked are given in the responses.

Reviewer 3 Report
Comments and Suggestions for Authors
This manuscript describes an experiment evaluating elevated zinc concentrations in the diet of dairy cows. I have 2 major concerns
1. The study seems to be underpowered. based on the number of tendencies discussed, it does not seem that 6 cows per treatment was enough. This cannot be corrected at this point, but should be discussed.
2. I think the conclusion about milk yield misrepresents the data since +Zn had lower milk yield in first 35 d. I think the conclusions need to be adjusted.
Specific comments can be found in the attached PDF.

Author Response
Thank you for your edits, comments, and suggestions. We revised the manuscript by addressing all your comments. Revised texts are highlighted in yellow in the revised manuscript. We transferred your comments from the PDF file to a Word file and responded to every item in the Word file uploaded herewith.
Reviewer 3
This manuscript describes an experiment evaluating elevated zinc concentrations in the diet of dairy cows. I have 2 major concerns
- The study seems to be underpowered. based on the number of tendencies discussed, it does not seem that 6 cows per treatment was enough. This cannot be corrected at this point, but should be discussed.
AU: Addressed in the revised manuscript. Please see the responses below
- I think the conclusion about milk yield misrepresents the data since +Zn had lower milk yield in first 35 d.
AU: Addressed in the revised manuscript. Please see the responses below
- I think the conclusions need to be adjusted.
AU: Addressed in the revised manuscript. Please see the responses below
Specific comments can be found in the attached PDF.
Reviewer’s comment and suggestion at line 3 of the previous submission
AU: correction was accepted
But milk yield was lower in first period so overall there was no difference in milk yield. Not sure if I would count that as an improvement. I dont think you measured milk yield long enough to make an conclusions about persistency.
AU: The focus on “persistency” was removed and the conclusion was revised as follows.
Despite the early DMI and MY reduction, the prolonged Zn-methionine supplementation at about 100 mg/kg of DM improved milk yield possibly as a result of improved udder health of dairy cows. (Lines 33-35)
Reviewer’s comment and suggestion at lines 53-54 of the previous submission
AU: correction was accepted
Reviewer’s comment and suggestion at line 68 of the previous submission
AU: correction was accepted
Even though the power analysis indicated 6 cows was enough, a lot of your variables are only tending toward significance. Rerun the power analysis using your data.
AU: A post hoc power analysis comparable to the initial analysis revealed sample sizes 7, 9, and 14 to capture the statistical significance (P < 0.05) of the observed effect sizes of SCC, SOD, and IgG, respectively. Therefore, future investigations equipped with improved statistical power would enable capturing the true effects of Zn-Met supplementation on the immune and antioxidant system-related traits of dairy cows. (Lines 382-386 of the revised manuscript)
With individual feeding the daily DMI was probably fluctuating a lot for individual cows from day to day. How did you achieve the 20 mg/kg DM?
AU: Depending on the average baseline DMI of +Zn cows added with a 10% safety margin (23.4 + 2.34 = 25.7 kg/d), 516 mg of total Zn-Met was top-dressed daily on the basal TMR delivered to every Zn+ cow to achieve the extra 20mg/kg of DM supplementation throughout the 70 d trial. When adjusted for the variability of DMI across the trial, the mean and the standard deviation (SD) of the Zn+Met supplementation were 20.9 and 3.5 mg/kg of DM, respectively. (Lines 88-93 in the revised manuscript)
Please report the total Zn in the CTL diet.
AU: The total Zn concentration was added to the table. The Zn from the basal feed ingredients were determined by using the nutrient composition tables (NRC, 2001). – Lines 167-168
Reviewer’s comment and suggestion at lines 216 of the previous submission
AU: correction was accepted (line 225)
Reviewer’s comment and suggestion at lines 237 of the previous submission
AU: correction was accepted (line 246)
This statement needs a citation. Also, should expand on this idea with more detail - the statement does not indicate how Zn and Mn interact or how it changes with stage of lactation.
AU: The statement was revised as follows
Zinc can modulate lactose synthesis depending on the Mn2+ concentration in mammary epithelial cells such that Zn can decrease lactose synthase activity at high intracellular Mn2+ concentrations [29]. Because Zn and manganese can share the same cell membrane transporters in the body [30], it can be speculated that the increased serum Zn of +Zn during the last than the first feeding phase competitively inhibited the Mn+2 uptake by mammary epithelial cells to increase lactose and thus milk yields. Nevertheless, data supporting such inhibition, particularly in mammary cells, are scares in published literature. (lines 254-261)
This sentence should start a new paragraph
AU: revised accordingly
Reviewer’s comment and suggestion at line 261 of the previous submission
AU: accepted the correction
Font size seems smaller here320-329
AU: Corrected
This seems unusual that Zn supplementation results in similar serum zinc. you mention that more zinc was going into milk, but the increase in milk zinc was less than for 70d where serum zinc was elevated. Are you correcting some type of deficiency during the first 35 d. This should be discussed more.
AU: The corresponding section was revised addressing the reviewer’s concerns as follows
The serum Zn concentration of +Zn being similar to that of CTL at d 35 despite the elevated Zn supply from the diet and the decreased milk Zn efflux was unexpected. One potential reason why the serum Zn concentration could reflect the Zn supplementation only at d 70 would be an increased immune activation [39] and thus, an increased Zn sequestration by the immune system [40] overriding the effects of dietary Zn supply and milk Zn efflux on blood Zn homeostasis in early-lactation compared to the mid-lactation in dairy cows. (Lines 325-331)
You are trying to make conclusions outside the power of your experiment. you cant do that.
AU: revised by addressing the following comments and suggestions from the reviewer as follows.
The results (Table 5) did not support any effects of +Zn on CAT, and MDA concentrations (P > 0.170). The Zn supplementation exhibited only a tendency to affect serum SOD concentrations in a time-dependent manner as indicated by the treatment × time interaction (P = 0.09). (Lines 369-372)
Reviewer’s comment or suggestion at line 355 of the previous submission
AU: Corrected
Reviewer’s comment or suggestion at line 359 of the previous submission
AU: Corrected
Line 370: This makes me think that you are going to discuss in-depth the activity of enzymes. Do you really mean further research should investigate activity of enzymes?
AU: revised as follows
Future investigations taking comprehensive approaches to determine the activity of different SOD species [53] and the activity of enzymes such as glutathione peroxidase that oxidizes GSH to GSSG [10, 11] may help draw robust conclusions about the effects of Zn supplementation on the degree of oxidative stress, antioxidant capacity or both in dairy cows. A post hoc power analysis comparable to the initial analysis revealed sample sizes 7, 9, and 14 to capture the statistical significance (P < 0.05) of the observed effect sizes of SCC, SOD, and IgG, respectively. Therefore, future investigations equipped with improved statistical power would enable capturing the true effects of Zn-Met supplementation on the immune and antioxidant system-related traits of dairy cows. (Lines 377-381)
Line 375-376: I do not think this is a valid interpretation of the data since milk yield was lower in first 35 d.
AU: Revised as follows
The results highlight that elevating dietary Zn-Met supplementation from 76.0 to 97.0±2.5 mg/kg of DM over 70 d decreased milk yield during the first half of the trial but was associated with increased milk yields during the second half despite DMI continued to be lowered by about 1.0 kg/d throughout the trial. (Lines 388-391).
Round 2
Reviewer 2 Report
Comments and Suggestions for Authors
Thank you very much for your corrections - I am satisfied with your changes.